TECHNICAL RELEASE

# PhysiPKPD: A pharmacokinetics and pharmacodynamics module for PhysiCell

Daniel Bergman[1],[*], Lauren Marazzi[2],[†], Mukti Chowkwale[3],[†],
Deepa Maheshvare M[4],[†], Supriya Bidanta[5],[†], Tarunendu Mapder[6] and
Jialun Li[7]

1 Department of Mathematics, University of Michigan, Ann Arbor, MI 48109, USA
2 Center for Quantitative Medicine, University of Connecticut School of Medicine, Farmington, CT 06030, USA
3 Department of Biomedical Engineering, University of Virginia, Charlottesville, VA 22908, USA
4 Department of Computational and Data Sciences, Indian Institute of Science, Bengaluru 560012, India
5 Indiana University, Bloomington, IN 47405, USA
6 Quantitative Systems Pharmacology, Bristol-Myers Squibb, Princeton Lawrenceville, NJ, USA
7 McGill University, Montréal, QC H3A 0G4, Canada

## ABSTRACT

Pharmacokinetics and pharmacodynamics (PKPD) are key considerations in any study of molecular therapies. It is thus imperative to factor their effects into any *in silico* model of biological tissue involving such therapies. Furthermore, creating a standardized and flexible framework will benefit the community by increasing access to such modules and enhancing their communicability. PhysiCell is an open-source physics-based cell simulator, i.e., a platform for modeling biological tissue, that is quickly being adopted and utilized by the mathematical biology community. We present here PhysiPKPD, an open-source PhysiCell-based package that allows users to include PKPD in PhysiCell models.

**Availability & Implementation:** The source code for PhysiPKPD is located here: https://github.com/drbergman/PhysiPKPD.

**Submitted:** 15 September 2022

* Corresponding author. E-mail: bergmand@umich.edu

† Contributed equally.

Preprint submitted at https://doi.org/10.1101/2022.09.12.507681

**Subjects** Software and Workflows, Cell Biology, Systems Biology

## STATEMENT OF NEED

Agent-based modeling has become a common tool for research in systems biology. However, it is still a relatively new tool and the community remains in need of standardized models that promote accessibility and extensibility. The PhysiCell platform is one such effort to achieve this [1, 2]. It is open-source, actively maintained, and has taken significant steps towards being widely adopted, as evidenced by the release of PhysiBoSS [3], an independently-developed integration of PhysiCell with intracellular signaling using Boolean modeling. New add-ons to PhysiCell will further broaden its appeal and facilitate the use of standards that will achieve the aforementioned community goals of accessibility and extensibility.

Many agent-based models (ABMs) have been developed to study diseases, including cancer [4–9], COVID-19 [10], tuberculosis [11, 12], and more, with a core purpose being to determine the best means to treat a patient with the given disease. This involves modeling therapeutics alongside the cells in a given microenvironment. The processes that determine

the effect of a given drug are commonly broken down into pharmacokinetics, or PK, and pharmacodynamics, or PD. Together, they are often called PKPD. Pharmacokinetics describes how a drug is transported throughout the body to get to the cells it acts on, i.e., the drug exposure. Pharmacodynamics describes how the drug then acts on those cells, i.e., the drug response.

The dynamics involved in PKPD are generalizable to many different drugs acting in different ways. This makes it an appealing target for a single module that can flexibly handle these processes on a platform such as PhysiCell. Features on both the PK side – such as dosing schedules, loading doses, elimination, and distribution rates – and the PD side – such as mechanisms of action (MOA), effect, the half maximal effective concentration or EC50, and hill coefficients – are amenable to this level of abstraction.

We present here PhysiPKPD, a standardized framework to incorporate these PKPD processes in PhysiCell. We also provide several examples demonstrating PhysiPKPD and providing users with two template projects to aid in the model-building process. In this first version, we provide three options for modeling pharmacokinetics: (1) one-compartment models, (2) two-compartment models, and (3) the ability to supply a Systems Biology Markup Language (SBML)-defined model. For options (1) and (2), the equations must be linear. On the PD side, substrates cause damage to cells based on the internalized substrate. As this damage accumulates within a cell, the MOA-associated rate parameter(s) tend towards a user-defined saturation rate. While we expect most users to add both PK and PD for a given substrate, they can be added independently if desired. For substrates that do have PD, these are set for each cell type in the model.

## IMPLEMENTATION

We provide two ways to create and run PhysiPKPD models, shown in Figure 1. First, the sample projects that come with PhysiPKPD demonstrate the MOAs implemented in PhysiPKPD (Figure 1A). Second, the user can use the template projects to jumpstart the model-building process with the full range of possible parameters and code present throughout the PhysiCell repository (Figure 1B). We now explain how PhysiPKPD achieves these dynamics. In what follows, S will stand for the name of a substrate, C for the name of a cell type, and X for an MOA.

### Pharmacokinetics

For pharmacokinetics, the user must first specify which substrates in the model follow a PK model. This is done by adding these substrate names to the comma-separated string user parameter PKPD_pk_substrate_names.

#### PK models

For each of these substrates, the user should specify a PK model (see Table 1). If this is not specified, PhysiPKPD will attempt to use a two-compartment model. In addition, the user should specify a Biot number, which PhysiPKPD uses to set the ratio of perivascular substrate concentration to that in the blood vessel [13]. PhysiPKPD will default to a value of 1, indicating equal concentrations inside and outside the capillary walls.



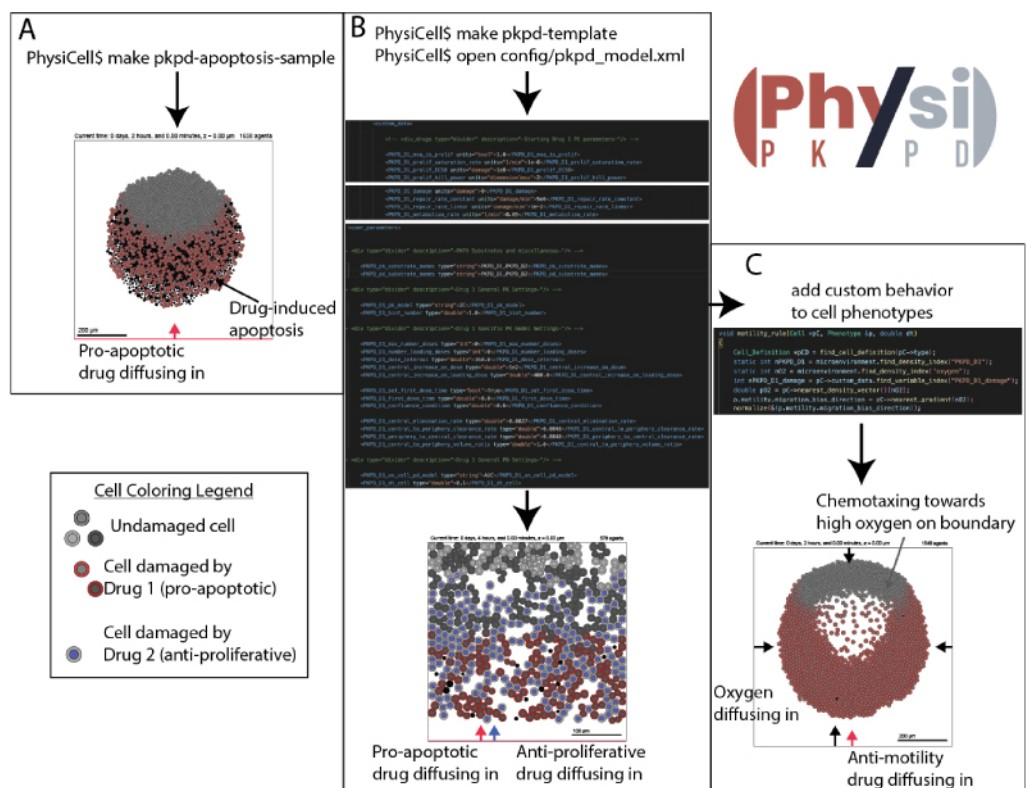

**Figure 1.** How to implement PhysiPKPD. (A) Using the sample projects that come with PhysiPKPD. (B) Using a template project and setting PKPD parameters in the configuration file. (C) Editing the C++ code in custom modules to further refine the model, e.g.making cell chemotax along an oxygen gradient.

**Table 1.** PK model specifications. The specification column indicates the string value to be used in the configuration file for the parameter `S_pk_model`.

| Model | Description | Specification |
|---|---|---|
| One-compartment | Circulation compartment with linear elimination | `1C` |
| Two-compartment | Circulation and peripheral compartments with linear clearance rates | `2C` |
| SBML-defined | Any SBML-defined model with one species named `circulation_concentration` | `SBML` |

For example, add the following to User Parameters in the configuration file: `<S_pk_model type="string">2C</S_pk_model>`.

*One-compartment models.* The PhysiPKPD one-compartment model obeys the differential equation

$$\frac{\mathrm{d}C}{\mathrm{d}t} = -\lambda C \tag{1}$$

where $C$ represents the circulation concentration of the substrate and $\lambda$ is the elimination rate (see Table 2). PhysiPKPD uses the value of $C$ to update the Dirichlet conditions in the PhysiCell microenvironment after multiplying by the Biot number.



**Table 2.** PK parameters for 1- and 2-compartment models.

| Parameter | Description | Type | Units |
|---|---|---|---|
| S_central_elimination_rate ($\lambda$) | Rate of elimination of the drug in systemic circulation | double | min$^{-1}$ |
| S_central_to_periphery_volume_ratio$^*$ ($R$) | Ratio of volumes of central and peripheral compartments to conserve mass of drug during distribution and redistribution | double | None |
| S_central_to_periphery_clearance_rate$^*$ ($k_{12}$) | Rate of change in concentration in central compartment due to distribution | double | min$^{-1}$ |
| S_periphery_to_central_clearance_rate$^*$ ($k_{21}$) | Rate of change in concentration in periphery compartment due to redistribution | double | min$^{-1}$ |

Replace S with the name of the substrate.
*This parameter is not necessary for 1-compartment models.

*Two-compartment models.* The PhysiPKPD two-compartment model obeys the differential equation

$$
\frac{\mathrm{d}C}{\mathrm{d}t} = \frac{k_{21}}{R}P - k_{12}C - \lambda C
$$
$$
\frac{\mathrm{d}P}{\mathrm{d}t} = k_{12}RC - k_{21}P.
$$

(2)

Here, $C$ and $\lambda$ are as above. $P$ is the periphery concentration. The parameters $k_{12}$ and $k_{21}$ are the intercompartmental clearance rates and $R$ is the ratio of the volumes of the central and peripheral compartments (see Table 2). PhysiPKPD uses the value of $C$ to update the Dirichlet conditions in the PhysiCell microenvironment after multiplying by the Biot number. The inclusion of the periphery compartment allows for biphasic elimination in the central compartment.

*SBML-defined models.* If the above two models are inadequate for a user's purposes, an SBML file can be used to specify a PK model. We have used Copasi (RRID:SCR_014260) [14] to build such models, but any program that outputs an SBML file will work, e.g., sbmlutils [15]. PhysiPKPD will use the first state variable of the model to update the Dirichlet conditions in the PhysiCell microenvironment after multiplying by the Biot number.

### Dosing schedules

For the one- and two-compartment models, all parameters and dosing events must be specified within the configuration file (see Table 3). For any missing parameters, PhysiPKPD will issue warnings and use default values where it can, and it will throw errors where it must. All substrates are assumed to be given intravenously so that the concentration in the central compartment has a one-time increase upon dosing, S_central_increase_on_dose. These doses are given at regular intervals, S_dose_interval, until either the simulation ends or until the maximum number of doses has been administered. The first dose can be given at a fixed time or based on the confluence in 2D in the entire rectangular microenvironment. A loading dose can also be set with a fixed number of doses. Future releases can include other methods of administration, e.g., oral, and finer-grained control, such as more sophisticated timings for doses, e.g., M-F dosing.

For SBML-defined models, the user must include dosing events in the SBML file itself. In Copasi, this can be done by creating Events that increase the concentration of a compartment(s) at certain times. In the future, we hope to allow the user to specify a CSV file with the dosing times and amounts for a substrate along with an SBML-defined system

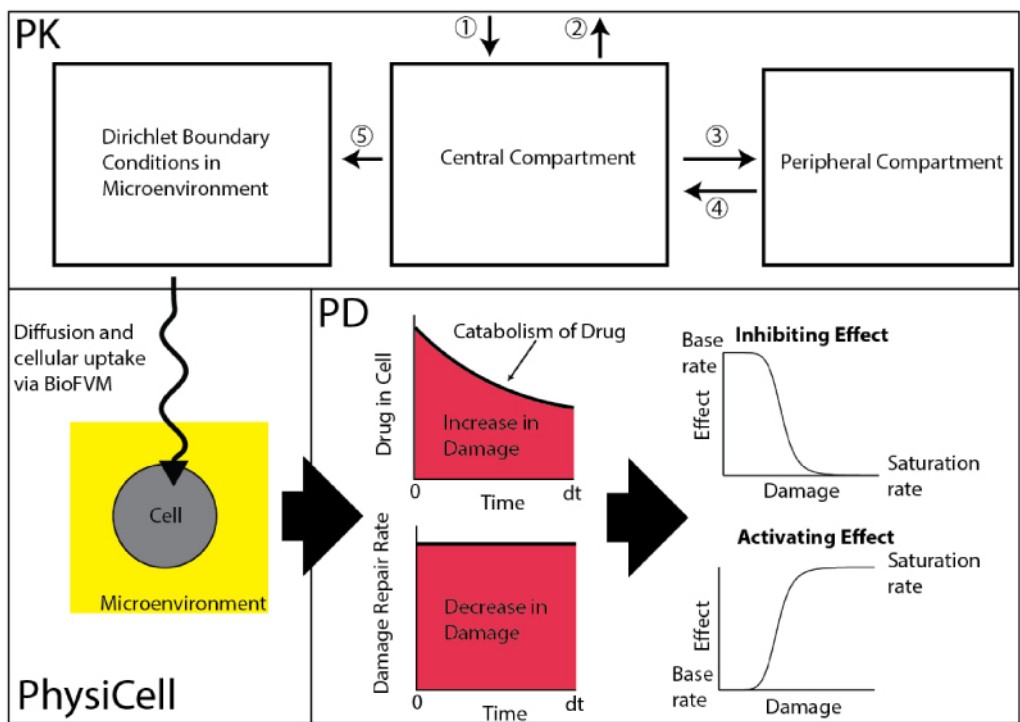

**Figure 2.** Integration of the PK and PD modules with PhysiCell. PK: (1) Administration of the drug in the central compartment. (2) Elimination of the drug. (3) Distribution of the drug into a peripheral compartment. (4) Redistribution of the drug back into the central compartment. (5) Extravasation of the drug into the microenvironment. PhysiCell: Diffusion and cellular uptake of the drug in the microenvironment. PD: the drug causes damage to cells over time. Cells can catabolize the drug and repair the damage. The damage determines the effect of the drug on its MOA.

**Table 3.** Dosing parameters for 1- and 2-compartment models.

| Parameter | Description | Type | Units |
|---|---|---|---|
| S_max_number_doses | Max number of doses to administer, including loading | int | # |
| S_number_loading_doses | Of the max number of doses, number of which are loading doses | int | # |
| S_dose_interval | Time between consecutive doses of S | double | min |
| S_central_increase_on_dose | Increase to systemic circulation concentration of substrate when a new dose is given | double | a.u. |
| S_central_increase_on_loading_dose | Increase to systemic circulation concentration of substrate when a new loading dose is given | double | a.u. |
| S_set_first_dose_time | Whether the user will set the time of the first dose or rely on confluence to begin dosing | bool | None |
| S_first_dose_time | Time of first dose (if setting) | double | min |
| S_confluence_condition | Proportion of microenvironment filled with cells when first dose given (if not setting first dose time) | double | Proportion |

Replace S with the name of the substrate.
a.u. = arbitrary units.

of ordinary differential equations (without dosing events) that PhysiPKPD then combines into a new SBML with dosing events.

**Table 4.** Damage accumulation model specifications. The specification column indicates the string value to be used in the configuration file for the parameter S_on_C_pd_model.

| Model | Description | Specification |
|---|---|---|
| Concentration-based | Damage accumulates in a cell based on the concentration of the internalized substrate | `AUC` |
| Amount-based | Damage accumulates in a cell based on the amount of the internalized substrate | `AUC_amount` |

For example, add the following to User Parameters in the configuration file: `<S_on_C_pd_model type="string">AUC</S_on_C_pd_model>`.

## Pharmacodynamics

Analogous to identifying substrates following a PK model, the user must specify which substrates follow a PD model. This is done by adding these substrate names to the comma-separated string user parameter `PKPD_pd_substrate_names`. While we expect this list to match that of `PKPD_pk_substrate_names` in most cases, that need not be true. For example, a substrate in the model that is produced by specific cells could result in pharmacodynamic effects on other cells. The processes of diffusion, degradation, and cellular production of such a substrate would be handled by PhysiCell and BioFVM [1].

### Damage accumulation

For each cell type affected by a particular substrate, a damage-accumulation model should be specified. If this is not specified, PhysiPKPD will default to the concentration-based model. The two options are the concentration-based model and the amount-based model (see Table 4). Both of these models use the area under the curve (AUC) of the internalized substrate to determine the current damage by the substrate on a cell. Future releases of PhysiPKPD may allow for SBML-defined damage accumulation functions. The experienced PhysiCell user can opt to use the already-implemented Intracellular features of PhysiCell to achieve this.

For either model, let $A$ represent the internalized substrate as either concentration or amount. Cells metabolize the substrate at a rate $m$. The damage accumulated by cells is proportional to the internalized substrate. This damage is repaired at the linear rate $r_1$ and the constant rate $r_0$. As damage is an abstract quantity, the proportionality constant for the internalized substrate causing damage is set to 1. All three of these parameters (Table 5) must be included in the custom data for C, the cell type affected by substrate S. The differential equations are then given by

$$\frac{\mathrm{d}A}{\mathrm{d}t} = -mA$$
$$\frac{\mathrm{d}D}{\mathrm{d}t} = A - r_1 D - r_0. \tag{3}$$

This equation is solved by default at the mechanics time step. The user can change this by setting the parameter S_dt_C. PhysiPKPD uses the analytic solution to solve these dynamics. By default, PhysiPKPD does this by pre-computing the relevant quantities. However, in cases where the above parameters can vary by cell type, the user should set the parameter `PKPD_precompute_all_pd_quantities` to false. Finer-grained control can be set by S_precompute_pd_for_C, which overrides `PKPD_precompute_all_pd_quantities` on a case-by-case basis.

**Table 5.** Damage accumulation parameters.

| Parameter | Description | Units |
|---|---|---|
| S_metabolism_rate ($m$) | Metabolism rate of internalized substrate | $\text{min}^{-1}$ |
| S_repair_rate_constant ($r_0$) | Zero-order repair rate of damage | damage/min |
| S_repair_rate_linear ($r_1$) | First-order repair rate of damage | $\text{min}^{-1}$ |

Set all in Custom Data of the cell definition of C.

**Table 6.** Cell effect parameters.

| Parameter | Description | Units |
|---|---|---|
| S_moa_is_X | Identifies whether the MOA of S on C is X | None (Values > 0.5 trigger the MOA) |
| S_X_saturation_rate | Limiting rate of X as damage from S grows towards infinity | $\text{min}^{-1}$ |
| S_X_EC50 | Damage from S at which the rate of X is halfway between the base and saturation rates | damage |
| S_X_hill_power | Hill power | None |

Replace X with one of the following: prolif, apop, necrosis, or motility.

### Cell effect parameters

To apply the desired MOA of S on C, the damage variable S_damage is used as the input to a Hill-type function. The user must specify three parameters for this Hill-type function in addition to identifying this MOA (see Table 6). The four MOAs currently implemented in PhysiPKPD are proliferation (prolif), apoptosis (apop), necrosis (necrosis), and motility (motility). Replace X in Table 6 with the parenthetical name of the desired MOA. By default, all MOAs are assumed off, so including S_moa_is_X in the custom data is only necessary if set to true. Note that custom data in PhysiCell must be of type double, so "true" means a value > 0.5.

*Multiply-targeted MOAs.* In the case that two or more substrates have the same MOA on a given cell type, we compute the drug effects as multipliers or factors. That is, the algorithm computes the saturation factor, $f_{\text{sat}}$, based on the user-supplied saturation rate and uses the damage to compute a factor, $f$, between 1 (no change) and the saturation factor (max change):

$$f = 1 + \frac{(f_{\text{sat}} - 1)(D/EC50)^n}{1 + (D/EC50)^n}, \quad f_{\text{sat}} = \frac{\text{saturation rate}}{\text{base rate}}. \tag{4}$$

When for example, multiple substrates affect proliferation, the factors for each substrate are multiplied to give the final factor for the cell. In the case of necrosis, the base rate for necrosis is often set to 0, which causes problems in computing $f_{\text{sat}}$. Thus, we instead add the effects of multiple substrates targeting necrosis.

*Resetting to base rates.* The implementation of PD is often done in the context of other dynamics. Typical examples used in the PhysiCell community include extracellular oxygen concentration and overcrowding (sometimes determined by simple pressure). PhysiPKPD therefore does not reset the MOA-targeted rates to their base value before applying the output of the Hill-type function. Thus, it is incumbent on the user to make sure the effects on these rates do not stack each time these effects are computed. See the custom.cpp files in any of the PhysiPKPD sample projects or the template projects to see how this can be done.

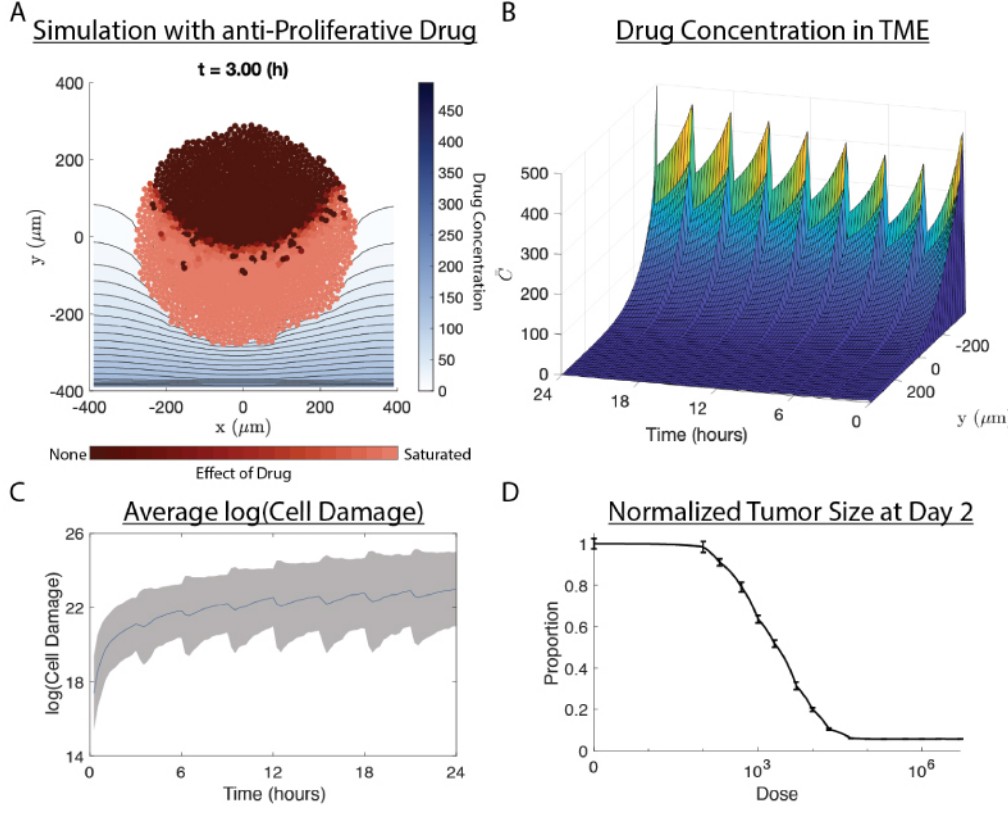

**Figure 3.** Results of the PhysiPKPD `pkpd-proliferation-sample` example. An anti-proliferative drug enters the microenvi-
ronment from the bottom boundary ($y = -400$ μm) and diffuses up towards the tumor. (A) Snapshot of the simulation after 3 h. Drug concentration shown by contour plot. Cells colored by the current effect of drug damage on their proliferation rates. (B) Drug concentration in the microenvironment over time. The average is taken over the $x$ dimension. (C) Average logarithm of cell damage over time. The average is taken over cells with positive damage only. The gray area represents +/− the SD of the log-transformed damage. (D) IC50 curve generated from running this example with just one dose of the drug given at time $t = 0$ over 16 different dosesincluding a control with no drug. The mean and SD for each dose were computed from 5 samples. The tumor size at $t = 48$ h was relative to the control shown. Note: analysis and initial plotting of (A–D) were done in MATLAB 2022a (RRID:SCR_001622) using the output from the PhysiCell simulation. The code that produces these figures is not part of PhysiPKPD.

## Example results

We provide several examples of implementations of PhysiPKPD: one for each MOA, one combination therapy with one anti-proliferative drug and one pro-apoptotic drug (Figure 1A), and one using a confluence condition to start dosing. Follow the README.md file instructions to get PhysiPKPD set up in the root PhysiCell directory. The output will go to the output folder. There are many parameters that can be changed to explore various behaviors, even restricted to just one MOA. See Tables 2, 3, 5, and 6 for the parameters and their descriptions unique to PhysiPKPD. In Figure 3, we show the results from the anti-proliferative drug sample project, including creating an IC50 curve by varying the dose over several orders of magnitude.

## DISCUSSION

A PKPD model that is ready to use "out of the box" with PhysiCell will greatly benefit those seeking to use PhysiCell as a platform for agent-based modeling. It will also create a standard that can be used by any modeler looking to incorporate PKPD into their PhysiCell models.

PhysiPKPD provides exactly this functionality. We designed PhysiPKPD so that the user can largely forego editing the C++ code, instead selecting PK/PD models and setting parameter values in the PhysiCell configuration file. By integrating with the native substrate class in PhysiCell, we can provide all these dynamics without introducing additional dependencies. When the PK models are the standard one- or two-compartment models, we use the analytic solution to provide fast and accurate updates to the pharmacokinetics. More importantly, the PD model also yields an analytic solution that we use to efficiently solve these dynamics, particularly compared to using numerical methods.

The flexibility of PhysiPKPD allows the user to readily add as many drugs as desired and have them affect the various cell types in unique ways. Each drug can follow its own PK model, and if the one- and two-compartment models are not sufficient, an SBML-defined PK model can take their place. Each drug can affect each cell type in a unique way, combining any number of PhysiPKPD's MOAs. By using the `damage_coloring` function of PhysiPKPD, a user can easily observe the extent to which up to two drugs are affecting a given cell type. This helps in narrowing down a parameter regime that results in observed and/or desired drug exposure and response.

We have also aimed to make PhysiPKPD easy for others to pick up and use. The samples provided are a great place to start learning these capabilities of PhysiPKPD. For those looking to build a new project with PKPD included from the beginning, we provide two sample projects to help the user get started: one with and one without SBML support for a PK model. Finally, adding PK and/or PD to a substrate in a current PhysiCell project is simply a matter of adding the correct header files, updating the Makefile, and adding the parameters to the configuration file.

There are many improvements and additions we hope to make to PhysiPKPD in the near future. With regards to PK, we hope to make it easier to include dosing events in an SBML file by allowing the user to supply a PK model and a CSV with the dosing information. With regards to PD, there are many features that can be added to allow for greater flexibility in terms of mechanisms of action, effect models, and integration with intracellular signaling. We have so far assumed that the AUC of a substrate inside a given cell, what we called damage, can be used to determine the effect of a drug on that cell. This is similar to irreversible effects models that use the AUC of the drug concentration to determine the response [16]. Many other effects models have been used, including simple direct effect, indirect response, and signal transduction models [17]. Recent work has used Bayesian inference to determine the distribution of delay times in response to therapeutic agents [18], making stochastic effects models an appealing next step as well. This could be added directly into previously-studied intracellular models or be used as a means to coarse-grain such complex models while still allowing for heterogeneity within cell types.

We are also working with the makers of PhysiCell to further simplify and standardize the integration of PhysiPKPD into PhysiCell. A key goal of this partnership will be to allow the user to include PKPD while only needing to specify the relevant parameters and without directly interacting with the C++ code.

We look forward to continuing to develop this tool as the community uses it and seeks new features.

## AVAILABILITY OF SOURCE CODE AND REQUIREMENTS

- Project name: PhysiPKPD
- Project home page: https://github.com/drbergman/PhysiPKPD
- Operating system(s): Platform independent
- Programming language: C++
- Other requirements: PhysiCell 1.10.4 or higher, libRoadRunner
- License: e.g. BSD 3-clause license
- RRID:SCR_022941

## DATA AVAILABILITY

Snapshots of the code and underlying data is available from the GigaScience GigaDB repository [19].

## DECLARATIONS

### List of abbreviations

ABM: agent-based model; AUC: area under the curve; MOA: mechanism-of-action; PD: pharmacodynamics; PK: pharamcokinetics; SBML: Systems Biology Markup Language.

### Ethical Approval

Not applicable.

### Competing Interests

TM is an employee of Bristol-Myers Squibb.

### Funding

None.

### Author's Contributions

Conceptualization: DB, LM, MC, TM; Methodology: DB; Software: DB, LM, MC, DM, SB; Validation: DB, LM, MC, DM, SB, TM, JL; Visualization: DB; Writing – original draft: DB; Writing – review & editing: DB, LM, MC, DM, SB, TM, JL.

### Acknowledgements

We would like to thank the PhysiCell team for all the help they have provided along the way. In particular, we would like to thank Paul Macklin, Randy Heiland, John Metzcar and Furkan Kurtoglu.

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
