## [Editor Report]

Comments to the AuthorPlease finalize the GigaDB entry with the curators and we'll complete the proofing and switch the paper live.

---

## [Reviewer Report]

Reviewer name and names of any other individual's who aided in reviewerJeffrey WestDo you understand and agree to our policy of having open and named reviews, and having your review included with the published manuscript. (If no, please inform the editor that you cannot review this manuscript.)YesIs the language of sufficient quality?YesPlease add additional comments on language quality to clarify if neededIs there a clear statement of need explaining what problems the software is designed to solve and who the target audience is? YesAdditional CommentsIs the source code available, and has an appropriate Open Source Initiative license <a href="https://opensource.org/licenses" target="_blank">(https://opensource.org/licenses)</a> been assigned to the code?YesAdditional CommentsAs Open Source Software are there guidelines on how to contribute, report issues or seek support on the code?YesAdditional CommentsIs the code executable?Unable to testAdditional CommentsIs installation/deployment sufficiently outlined in the paper and documentation, and does it proceed as outlined?Unable to testAdditional CommentsIs the documentation provided clear and user friendly?YesAdditional CommentsIs there enough clear information in the documentation to install, run and test this tool, including information on where to seek help if required?YesAdditional CommentsIs there a clearly-stated list of dependencies, and is the core functionality of the software documented to a satisfactory level?YesAdditional CommentsHave any claims of performance been sufficiently tested and compared to other commonly-used packages? YesAdditional CommentsIs test data available, either included with the submission or openly available via cited third party sources (e.g. accession numbers, data DOIs)?YesAdditional CommentsAre there (ideally real world) examples demonstrating use of the software? YesAdditional CommentsIs automated testing used or are there manual steps described so that the functionality of the software can be verified?YesAdditional CommentsAny Additional Overall Comments to the AuthorThis is a very nice & useful extension to PhysiCell, in order to model PK/PD dynamics in agent-based simulations. Overall, the description of the software is good and easy to follow, but I offer a few suggestions for clarity:  1. In "Statement of Need" -- the phrase "how much gets to the cells and what they then do to the cells" is vague and casual -- maybe use standard terms like drug exposure & response to describe PK/PD relationships 2. Final sentence in "Statement of Need" that says "Substrates can target any cell type with PD dynamics" -- can you elaborate? Does this indicate that every cell type can have unique PD dynamics? 3. In "Implementation" authors refer to Figure 2A and 2B but figure 2 only has one panel -- perhaps this should be figure 1A/B? 4. In "Pharmacodynamics" -- "the list of PK substrates and the list of PDsubstrates need not have any relationship" -- this is slightly confusing. I assume that every substrate can have associated PK dynamics without having an PD dynamic, but is the opposite true? If so, how what is the drug dispersal / decay rate? 5. Finally, the discussion section is focused mainly on future steps. I think it would be helpful for the discussion to focus more on current advantages and functionality. This is the publication record for this software, and as is often the case, future steps may be subject to change.RecommendationMinor Revisions

---

## [Reviewer Report]

Reviewer name and names of any other individual's who aided in reviewerBoris AguilarDo you understand and agree to our policy of having open and named reviews, and having your review included with the published manuscript. (If no, please inform the editor that you cannot review this manuscript.)YesIs the language of sufficient quality?YesPlease add additional comments on language quality to clarify if neededIs there a clear statement of need explaining what problems the software is designed to solve and who the target audience is? YesAdditional CommentsIs the source code available, and has an appropriate Open Source Initiative license <a href="https://opensource.org/licenses" target="_blank">(https://opensource.org/licenses)</a> been assigned to the code?YesAdditional CommentsAs Open Source Software are there guidelines on how to contribute, report issues or seek support on the code?NoAdditional CommentsIs the code executable?NoAdditional CommentsThis code can not be in an. executable form as is an extension to PhysiCellIs installation/deployment sufficiently outlined in the paper and documentation, and does it proceed as outlined?Unable to testAdditional CommentsI ma not familiar with running PhysiCellIs the documentation provided clear and user friendly?YesAdditional CommentsIs there enough clear information in the documentation to install, run and test this tool, including information on where to seek help if required?YesAdditional CommentsIs there a clearly-stated list of dependencies, and is the core functionality of the software documented to a satisfactory level?YesAdditional CommentsHave any claims of performance been sufficiently tested and compared to other commonly-used packages? NoAdditional CommentsAuthor claim this is the first time PKPD module has been added to PhysiCell. Is test data available, either included with the submission or openly available via cited third party sources (e.g. accession numbers, data DOIs)?YesAdditional CommentsAre there (ideally real world) examples demonstrating use of the software? YesAdditional CommentsIs automated testing used or are there manual steps described so that the functionality of the software can be verified?NoAdditional CommentsAny Additional Overall Comments to the Author- I think there is mistake in calling Figure 1 in Installation sections, should be Figure 1. - Reference to PhysiBoSS missing  - Figure 1 RecommendationMinor Revisions